# Closing Tobacco Treatment Gaps for Rural Populations: The Role of Clinic-Based Pharmacists at a Federally Qualified Health Center

**DOI:** 10.3390/pharmacy13010010

**Published:** 2025-01-26

**Authors:** Lavinia Salama, Karen Suchanek Hudmon, Leena Myran, Nervana Elkhadragy

**Affiliations:** 1School of Pharmacy, University of Wyoming, Laramie, WY 82071, USA; lmyran@uwyo.edu (L.M.); nelkhadr@uwyo.edu (N.E.); 2University of Wyoming Family Medicine Residency at Cheyenne, Cheyenne, WY 82001, USA; 3Department of Pharmacy Practice, College of Pharmacy, Purdue University, West Lafayette, IN 47907, USA; khudmon@purdue.edu

**Keywords:** tobacco cessation, smoking cessation, pharmacist counseling, rural healthcare, federally qualified health center (FQHC), patient engagement, advocacy, collaborative practice agreement, pharmacist-led interventions, rural communities

## Abstract

Pharmacists are often the first point of contact for healthcare advice in rural communities, where access to healthcare is limited. Tobacco cessation rates improve with counseling from a pharmacist, and in many states, pharmacists can now prescribe medications for quitting. This study aimed to explore smoking behavior and cessation motivations among patients at a Federally Qualified Health Center (FQHC) clinic in rural Wyoming, estimate the prevalence of tobacco-related interventions by clinic staff, and assess patients’ interest in engaging in pharmacist-led cessation programs. A cross-sectional survey was administered over three months to clinic patients who self-identified as current tobacco users. Survey items assessed sociodemographics, tobacco use and vaping behaviors, previous cessation advice from pharmacists, and interest in pharmacist-led support for quitting. Of 63 respondents, 57 (90.5%) reported current tobacco use. Most were ready to quit within the next month (43.9%) or the next six months (33.3%), and 26.3% had previously received advice about quitting from a pharmacist. Most (59.6%) expressed interest in establishing care with a pharmacist for cessation support, and 45.3% requested to be contacted by a pharmacist for assistance with quitting. Interest did not differ by gender or age. The results indicate that rural patients are interested in engaging with pharmacists for quitting. Further research is needed to determine how pharmacy-led programs can complement existing healthcare resources to improve access to cessation support in underserved areas.

## 1. Introduction

Tobacco use is a leading contributor to avoidable disease, disability, and death in the United States [1], yet in 2021, approximately 46 million adults in the U.S. (18.7%) reported being a current user of one or more tobacco products [2]. Despite proven methods, most individuals attempt to quit on their own, without the use of medications and without professional assistance [3]. Rural states, including Wyoming, often rank among the lowest in the nation in terms of tobacco users receiving cessation advice from medical professionals [4]. The low-frequency tobacco cessation service provision in underserved communities is likely impacted by factors such as inadequate insurance coverage, challenges with patient transportation in rural regions, and disparities in cessation service coverage across different insurance plans [5]. In Wyoming, an important step towards addressing this gap in care would be granting expanded authority for pharmacists to prescribe tobacco cessation medications [6]—similar to pharmacists in at least 17 other states [7]. This would not only render evidence-based cessation assistance more accessible, but it would also help to alleviate the burden on primary care providers [6,8]. Furthermore, the anticipated primary care physician shortage, which is projected to worsen by 2033, will likely exacerbate these issues and lead to fewer patients receiving tobacco cessation counseling [8]. The objective of this study was to evaluate patients’ smoking behaviors, readiness to quit, and interest in pharmacist-led tobacco cessation services, while assessing the current provision of tobacco-related interventions by clinic staff, to inform legislative efforts for expanding pharmacist prescribing authority in Wyoming.

The published literature supports the impact of pharmacists in providing tobacco cessation interventions; however, additional research is needed to characterize the interest of rurally-located patients’ interest in quitting smoking and interest in establishing tobacco cessation assistance with a pharmacist [9,10,11]. Pharmacists are ideally positioned to deliver these interventions, especially in rural areas, due to their accessibility, trustworthiness, and expertise [12]. With most Americans living within five miles of a pharmacy, pharmacists represent an essential healthcare resource in regions with limited primary care providers [12,13]. Rural residents often exhibit high levels of trust in their local pharmacists, fostering open discussions about smoking cessation [12]. Additionally, pharmacists’ training in medication counseling and chronic disease management enables them to effectively support adherence to medication regimens. A recent qualitative study emphasized the critical role of community pharmacies in bridging the tobacco treatment gap, particularly for underserved populations [14]. The study demonstrated that pharmacists and pharmacy technicians, after undergoing tobacco cessation training, successfully implemented cessation services in community pharmacy settings [14]. Pharmacists’ ability to prescribe tobacco treatment medications underscores the potential for expanded pharmacy-based care models. Toward a goal of advancing legislation for prescribing in Wyoming, this study aimed to (a) gain a deeper understanding of patients’ smoking behavior and their desire to quit within a Federally Qualified Health Center (FQHC) clinic in rural Wyoming, (b) assess the current frequency of tobacco-related interventions provided by clinic staff, and (c) characterize patients’ interest in establishing tobacco cessation care with a pharmacist.

## 2. Materials and Methods

### 2.1. Survey Design

This survey-based research was conducted at The University of Wyoming Family Medicine Clinic at Cheyenne, an FQHC clinic specializing in family medicine and prioritizing screening and prevention for its diverse and largely under-resourced patient population. The study aimed to enroll 250 adult patients (≥18 years old) between 16 March and 30 June 2023. Data collection, facilitated through electronic (in Qualtrics) and paper surveys available in English and Spanish, focused on understanding smoking behavior and related factors among adult patients. Initially, a USD 5 gift card was provided for survey completion; however, to enhance participation, this was later replaced with gift card raffle awards in the amounts of USD 10, 50, and 100. The survey targeted individuals who self-identified as a current tobacco user or vaper when visiting the clinic waiting room, by answering the following question: Do you NOW smoke cigarettes, vape products, or use any other forms of tobacco?

### 2.2. Framework

Survey questions were developed based on the Theory of Planned Behavior (TPB), predicting individuals’ intention to engage in behavior within a specific context. In the context of facilitating tobacco cessation care with pharmacists, TPB suggests that favorable attitudes towards quitting, social norms endorsing cessation, and perceived behavioral control are associated with greater intention to quit [15,16]. These constructs guided the development of survey items, ensuring alignment with key factors influencing smoking cessation intention (Figure 1) [15,16].

### 2.3. Survey Development and Measures

The study authors crafted the survey items, tailoring questions to address the study objectives through the TPB. These questions were reviewed by two researchers with extensive expertise in behavioral theory and tobacco cessation research. The survey items were initially developed in English, and a consultant specializing in patient-friendly language ensured their appropriateness for a patient population with limited medical literacy. Subsequently, the survey was translated into Spanish by another consultant, followed by re-translation back into English. Finally, a comprehensive verification process was conducted by a final consultant to confirm alignment between the English and Spanish versions.

The survey was divided into three sections. In the first section, participants were queried about their current tobacco use behavior, including frequency and types of tobacco or vaping products used, average daily cigarette consumption, duration of use, and environmental factors such as living with individuals who smoke and smoking indoors. Those who had attempted to quit were further asked about the timing of their last quit attempt, use of cessation aids, and previous smoking replacement methods. Additionally, participants were asked about their intentions to quit and perceived challenges to quitting.

The second section focused on prior counseling received from clinic staff members, including primary care providers and pharmacists. Participants were asked about the frequency and duration of discussions about tobacco use with their primary care providers, awareness of available cessation resources, and interactions with pharmacists regarding tobacco use and cessation assistance. The interest in establishing care with pharmacists, perceptions of the convenience of telehealth appointments, and intentions to seek pharmacist assistance for smoking cessation were also assessed.

Finally, survey items assessed gender, age, race/ethnicity, and willingness to be contacted by clinic staff for further information about quitting. The Appendix A includes a copy of the English and Spanish surveys.

Using IBM SPSS Statistics (Version 29), summary statistics were computed to characterize the patient population and survey responses. Chi-squared tests and t-tests were computed for group comparisons, as appropriate. All data were collected anonymously, and study procedures were approved by the University of Wyoming Institutional Review Board (IRB).

## 3. Results

### 3.1. Study Population

A total of 63 surveys were completed; of these, 57 (90.5%) respondents reported current tobacco use or vaping either every day or some days and were included in additional analyses pertaining to tobacco/vaping (Table 1). Most smoked cigarettes (*n* = 47; 82.5%), with an average of 16.7 per day (SD, 11.5), 19 reported vaping (33.3%), and 2 reported smokeless tobacco use (3.5%). Years of smoking, vaping, and tobacco use among 57 current users was less than one year (*n* = 1; 1.8%), 1 to 5 years (*n* = 7; 12.3%), 6 to 10 years (*n* = 8; 14.0%), 11–20 years (*n* = 14; 24.6%), more than 20 years (*n* = 26; 45.6%), and unsure (*n* = 1; 1.8%). More than half (*n* = 34; 59.6%) lived with someone who smokes, and 23 (40.4%) reported smoking inside their home. All surveys were completed in English.

### 3.2. Tobacco Use History and Interest in Quitting

Among 57 current tobacco users, 84.2% reported having tried to quit smoking during their lifetime, with just over half (52.1%) having attempted to quit in the past 12 months. Fifty percent had used a quit-smoking medication in the past. The most common was the nicotine patch (75.0%), followed by nicotine gum (70.8%), varenicline (41.7%), the nicotine lozenge (33.3%), and bupropion SR (29.2%) (categories are not mutually exclusive). Most participants were ready to quit in the next month (40.4%), some in the next 6 months (33.3%), and few (7.0%) were interested in quitting, but not in the next 6 months. The most prevalent concern about quitting was withdrawal symptoms (40.4%), followed by being around peers and family members who smoke (36.8%), weight gain (35.1%), and cost of quit-smoking medications (22.8%). Fifteen (26.3%) had previously been advised by a pharmacist to quit smoking, and 28.1% reported that a pharmacist had asked about their smoking. When asked about interest in establishing care with a pharmacist to help with quitting, 59.6% indicated that they were interested, and 24 respondents (45.3%) provided their information to be contacted by the pharmacy staff for assistance. Interest in receiving assistance from a pharmacist did not differ by gender (*p* = 0.79) or age (*p* = 0.51).

### 3.3. Counseling Received from Other Clinic Staff Members

Most respondents (75.8% of 63) reported seeing their primary care provider (PCP) at least once a year, and 85.5% indicated that the provider had asked if they smoked, vaped, or used other types of tobacco. Current tobacco users/vapers (*n* = 57) estimated that their PCP spent no time (24.6%), up to 3 min (40.4%), 3–5 min (22.8%), or more than 6 min (12.3%) talking about tobacco use. Most (78.9%) were aware that they could make an appointment to specifically discuss quitting smoking with their provider. Other clinic staff who had advised the respondents to quit smoking included nurses (57.9%).

### 3.4. The Role of Pharmacists in Quitting

Fewer than half of the patients were aware that pharmacists could assist with quitting, and more than half reported that they would be interested in establishing care with a pharmacist for stopping smoking (Figure 2). Among those who were interested (59.6%), 60.6% perceived it to be more convenient to establish care with a community pharmacist, and 38.9% agreed it would be more convenient to have appointments held online via telehealth (22.2% did not agree, and 38.9% had no preference). Half of current tobacco/vaping users intended to ask their pharmacist for help with quitting, and 35.2% intended to ask their pharmacist about the benefits and risks of smoking cessation medication use for their quit attempt. Six (11.1%) were interested in group sessions with a pharmacist and other tobacco users. Most (61.1%) believed that people important to them would be supportive of their receiving aid from a pharmacist to quit (7.4% disagreed; 31.5% were unsure).

## 4. Discussion

Pharmacists are well-positioned to provide personalized counseling, medication management, and ongoing support, leveraging their accessibility and expertise to address barriers to smoking cessation. A systematic review by Brown and colleagues highlighted the valuable role of community pharmacists in delivering public health services, particularly in smoking cessation, ref. [17] and other reviews have also demonstrated that pharmacists’ involvement in tobacco cessation enhances patient outcomes and health system efficiency [11,18,19]. A meta-analysis [20] and other studies of pharmacist-led interventions underscored pharmacists’ ability to significantly improve quit rates, reinforcing the impact of pharmacists’ direct engagement in smoking cessation efforts. Internationally, an intensive pharmacist-delivered program in Qatar was perceived as beneficial by both pharmacists and patients, indicating the approach’s adaptability across diverse healthcare systems [21]. Similarly, in Canada, a community-based pharmacist-led program successfully increased smoking cessation rates among patients awaiting elective joint replacement surgery, ref. [22] and a Canadian Agency for Drugs and Technologies in Health Rapid Response Report suggested that pharmacist-led interventions may not only yield higher quit rates but can also be cost-effective when compared to self-guided cessation methods [23]. Furthermore, other international experiences corroborate the impact of community pharmacists on smoking cessation. For instance, a large-scale study in Thailand demonstrated a 28.8% abstinence rate among smokers receiving pharmacist-led cessation services, with significant reductions in daily cigarette consumption, exhaled carbon monoxide levels, and improvements in pulmonary function [24]. Such evidence highlights the adaptability of pharmacist-led models across diverse healthcare systems and reinforces their essential role in expanding cessation support globally. By capitalizing on patient interest and leveraging pharmacist expertise, healthcare systems can expand the reach and impact of tobacco cessation efforts, especially in underserved rural areas where access to comprehensive cessation services is often limited [25,26].

The results of this study aligned with a survey conducted at a multi-site FQHC in Indiana, where nearly two thirds of participants expressed interest in utilizing a pharmacist’s expertise for quitting [27]. Similarly, in our study, almost half of the respondents requested to be contacted by pharmacy staff for cessation assistance, underscoring the potential for proactive outreach to increase engagement in cessation services. These findings reflect those of a rural Appalachia study, which found that recognizing pharmacists as smoking cessation resources could significantly increase quit attempts and cessation rates [25]. Our findings were consistent with previous research, demonstrating that pharmacist-led tobacco cessation clinics in rural locations have the potential to enhance access to evidence-based tobacco treatment.

While our findings suggest that prescriptive authority might not be essential for pharmacists to effectively facilitate smoking cessation—given that many patients in our study were already connected to healthcare providers who could authorize necessary prescriptions—they emphasize the important collaborative role that pharmacists play. This is particularly important in rural communities, where access to healthcare is limited, and establishing care with a community pharmacist may be more accessible and convenient for patients.

According to the Center of Disease Control (CDC), rural populations across the United States face significant health disparities, compounded by hospital closures and physician shortages [28]. Policymakers at federal and state levels have begun recognizing pharmacists as vital providers who can help bridge these care gaps [29]. Recent policy discussions have underscored the potential for pharmacists to practice at the top of their license by administering preventive services, helping patients manage chronic diseases, and providing more advanced counseling [29]. This shift could be especially impactful in remote communities, where pharmacists often function as the only local health professionals. For example, some proposals have considered reimbursing pharmacists in rural or underserved areas for patient care services rather than tying payment strictly to product dispensing, and such reforms could incentivize pharmacists to offer expanded cognitive services, ultimately lowering healthcare costs while improving health outcomes [29]. Embracing these models might help to alleviate the impact of provider shortages, enabling pharmacists to more fully leverage their advanced training in medication therapy management, public health, and preventive care, thus enhancing both health access and quality for rural populations.

Expanding pharmacists’ prescriptive authority, without requiring collaborative practice agreements (CPAs), could significantly enhance their ability to support cessation efforts for patients without established relationships with healthcare providers, thereby increasing access to evidence-based interventions [29,30,31]. Establishing CPAs between physicians and community pharmacists can be challenging due to barriers such as provider hesitancy, liability concerns, logistical hurdles, billing complexities, and insufficient resources or information to create and maintain these agreements [30,31]. Granting prescriptive authority directly to pharmacists could overcome these challenges, allowing them to take on a more active and autonomous role in comprehensive tobacco cessation strategies [29,30,31]. This approach would help bridge care gaps, expand access to evidence-based interventions, and improve public health outcomes, particularly in underserved areas facing physician shortages and healthcare disparities [28,29,30].

Rural populations in the United States face higher rates of tobacco use and related health disparities, including increased mortality from cancer, heart disease, and respiratory conditions, compared to urban populations [26]. With their accessibility and ability to prescribe tobacco cessation medications in some states, pharmacists are uniquely positioned to address healthcare gaps and enhance cessation efforts in these underserved areas [26]. Drawing from successful models from states such as New Mexico [32], where pharmacists’ prescriptive authority has improved access to cessation services, Wyoming policymakers are encouraged to adopt similar measures to reduce tobacco-related morbidity and mortality, and the data derived from this study will contribute toward this goal. Legislative support for pharmacist-led cessation services would not only help to alleviate the primary care physician shortage but also expand the healthcare workforce’s capacity to deliver essential preventive services, ensuring the continuity of care and improving population health outcomes [8].

One clear implication of this study, and others like it, is the need for sustained, specialized training in tobacco treatment for pharmacists [33]—especially those practicing in rural or underserved areas. A quasi-experimental study in Indonesia demonstrated that even a single-day workshop significantly enhanced pharmacists’ knowledge, perceived role, and self-efficacy in providing cessation counseling [34]. A working group within the American Association of Colleges of Pharmacy emphasized that all graduates should possess the evidence-based knowledge and skills to effectively intervene with patients who use tobacco [34]. Integrating tobacco content into pharmacy school curricula may help standardize these skills and ensure that graduates enter the workforce prepared to meet the unique challenges of rural healthcare [34,35]. Regular updates, refresher training, and culturally-tailored materials for post-graduates are also important, ensuring that the entire pharmacy workforce is able to support diverse patient populations in achieving successful quit attempts [36].

Despite the valuable insight provided by the study findings, it is important to acknowledge their limitations. One significant limitation is the small sample size, which may limit the generalizability of the findings to the broader population of tobacco users in rural Wyoming. Although our results align with a prior study with respect to FQHC patients’ interests in engaging with pharmacists for cessation [27], the study aimed to recruit 250 participants but was closed before reaching the target due to time constraints imposed by the funding timeline. The recruitment relied on clinic providers and staff to actively distribute the survey, a process that was inconsistently implemented at times due to understaffing. The small sample size also impacted the precision of the statistical analyses and reduced the statistical power to detect differences across demographic subgroups (e.g., race/ethnicity or age cohorts). To enhance participation, the method of incentives was changed mid-study, with the first 28 (49.1%) of the 57 respondents who reported smoking or vaping receiving a gift card, and the latter 29 (50.9%) respondents being entered into a gift card raffle. This might have introduced bias, although no statistically significant differences were identified for key sociodemographic variables or the number of cigarettes smoked per day. Additionally, the use of convenience sampling from a single clinic setting may have introduced selection bias, as patients who chose to participate may have differed systematically from those who did not. Finally, the reliance on self-reported data might have introduced recall bias or social desirability bias, affecting the accuracy of reported smoking behaviors and cessation intentions. Although a Spanish-language survey was offered, no responses were received in Spanish, highlighting the underrepresentation of Spanish-speaking patients and underscoring the need for more culturally-tailored recruitment strategies.

To enhance generalizability, future research should prioritize recruiting larger, more diverse samples across multiple clinic sites and practice settings and examining the relationships between patient characteristics and the interest in engaging with pharmacists for tobacco treatment. Additionally, research is needed to characterize perceptions of patients who have actually engaged with pharmacists for tobacco treatment services. Comparative analyses between pharmacist-led and other cessation models, as well as longitudinal studies examining long-term quit rates, would further clarify the effectiveness and sustainability of these interventions. Additionally, investigating cost-effectiveness and exploring innovative delivery methods (e.g., telehealth) could expand access to cessation services, especially in underserved rural communities.

## 5. Conclusions

Results of this study reveal significant interest in pharmacist-led tobacco cessation services among rural patients in Wyoming, suggesting an opportunity for pharmacists to play a key role in supporting cessation efforts. These data will be applied toward legislative efforts to permit Wyoming pharmacists to prescribe cessation medications, thereby potentially enhancing access to care for patients without established healthcare provider relationships. By continuing to explore targeted pharmacist-led interventions and collaborative efforts with policymakers, healthcare systems may improve cessation support in underserved communities and contribute to population health outcomes. Our findings provide a foundation for future studies and discussions around optimizing pharmacists’ roles in tobacco cessation.

## Figures and Tables

**Figure 1 pharmacy-13-00010-f001:**
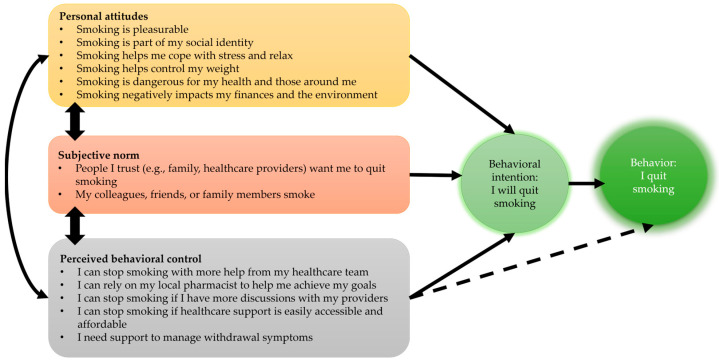
Theory of Planned Behavior: Determinants of Smoking Cessation. Solid lines: Represent the primary, well-supported pathways between variables according to the theory. Dashed lines: Represent potential relationships that may exist but require further research to confirm their strength and significance.

**Figure 2 pharmacy-13-00010-f002:**
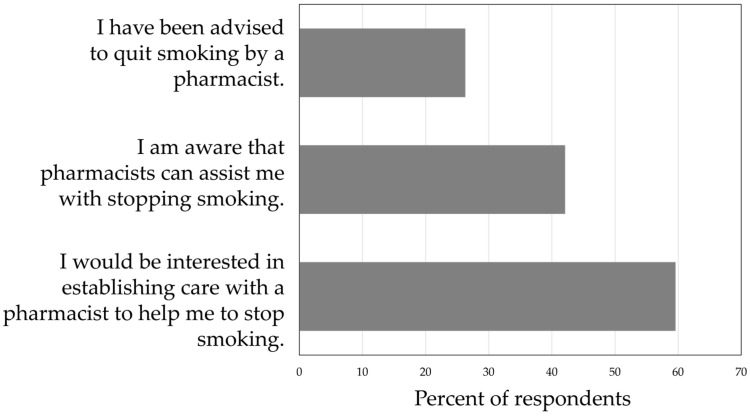
The pharmacists’ role in tobacco cessation: Participant responses (*n* = 57).

**Table 1 pharmacy-13-00010-t001:** Characteristics of current tobacco users (*n* = 57).

**Characteristic**	**Category**	**N (%)**
Gender	Female	27 (47.4)
	Male	26 (45.6)
	Prefer not to respond	4 (7.0)
Race/ethnicity	Caucasian/White	47 (82.5)
	Black/African American	5 (8.8)
	American Indiana/Alaska Native	2 (3.5)
	Other or prefer not to respond	3 (5.3)
Age	18–24	5 (8.8)
	25–34	13 (22.8)
	35–44	11 (19.3)
	45–54	11 (19.3)
	55–64	9 (15.8)
	65 or more	4 (7.0)
	Prefer not to respond	4 (7.0)
Frequency of seeing primary care provider	Less than once yearly	14 (24.6)
	Once a year	11 (19.3)
	Twice a year	7 (12.3)
	Three times a year	4 (7.0)
	More than three times a year	21 (36.8)

## Data Availability

The research data are not publicly available.

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
