# Peer review of "Closing Tobacco Treatment Gaps for Rural Populations: The Role of Clinic-Based Pharmacists at a Federally Qualified Health Center"

_pharmacy, 2025, doi:10.3390/pharmacy13010010_

Round 1
Reviewer 1 Report
Comments and Suggestions for Authors
This paper is very interesting, exploring a global issue like smoking cessation in underserved area.
Pharmacists can easily fill this gap.
The design of the study is very simple and it is repeatible in other context. This is an invitation to extend your study to other centers for a better performance of the statistical analysis (63 patients are very few)
We can take this study as a pilot one.
I didn't find any issue to address.
Author Response
Thank you for taking the time to review our paper and for your thoughtful feedback. We greatly appreciate your recognition of the study’s relevance in addressing smoking cessation in underserved areas and its potential for broader application. Thank you again for your valuable comments. As there were no specific issues raised, we have not made further revisions for this round.
Reviewer 2 Report
Comments and Suggestions for Authors
Dear Authors,
I appreciate you paper. In order to improve it please find below my comments:
1. it is not clear to me the rationale of your paper. The pharmacists' role is due to the proximity, or because pharmacists are trustable in rural areas, or other?
2. to improve the rationale I suggest to widen the literature review. I do not see a specific paragraph on that.
3. to strengthen your analysis and reinforce your conclusions, I suggest to explicit the research question in the first paragraph.
4. Even if you write that one limitation is the little sample, it would be useful to say something more about different cluster of interviewees.
5. I wonder if there are other similar researches in other countries or contexts
6. When you refer to further research, what do you really mean? only to have a larger sample, to promote comparative analysis, to add some other questions?
7. Can you deepen the effectiveness of rural pharmacists' role?
8. Do you see some implications on pharmacists training / education?
Author Response
Thank you for your detailed and constructive feedback. We have carefully considered your comments and have made the following revisions to improve the clarity and depth of our manuscript:
- Clarifying the Rationale
To address the rationale for pharmacists’ involvement, we have expanded the introduction to highlight their unique accessibility, trustworthiness, and expertise, with a focus on the importance of these characteristics in rural areas. We emphasized their ability to bridge gaps in care due to proximity to underserved populations and their training in medication management and counseling. Additionally, we included recent literature supporting the critical role of community pharmacists in tobacco cessation interventions. - Expanding the Literature Review
A dedicated section elaborating on the literature was added to the introduction. This includes a discussion of pharmacists’ roles in smoking cessation in various contexts, supported by recent qualitative and quantitative studies, to strengthen the theoretical foundation of our work. - Explicitly Stating the Research Question
The research question has been clearly articulated in the introduction to provide a focused framework for the study. Specifically, we outlined our objectives to evaluate patients’ smoking behaviors, readiness to quit, and interest in pharmacist-led cessation services, while also assessing current tobacco-related interventions at the clinic to inform legislative efforts. - Discussing Limitations and Clusters
We expanded on the limitations of the study, particularly regarding the small sample size and demographic diversity. We noted the underrepresentation of Spanish-speaking patients and the inability to perform robust subgroup analyses due to the limited sample size. These points are addressed in both the results and discussion sections to highlight areas for future improvement. - Including Similar Research from Other Contexts, including other countries
The discussion now includes examples of similar pharmacist-led interventions in diverse international and domestic settings, such as Canada, Qatar, Thailand, and other U.S.-based studies. These examples demonstrate the adaptability and potential effectiveness of pharmacist-led cessation models across different healthcare systems. - Clarifying Suggestions for Future Research
We elaborated on our recommendations for future research in the discussion, emphasizing the need for larger, more diverse samples, comparative analyses of different cessation models, cost-effectiveness studies, innovative delivery methods such as telehealth, and a need to explore the perceptions of patient who have actually engaged with pharmacists for assistance with quitting. - Deepening the Discussion of Pharmacists’ Role
Additional context on the potential of rural pharmacists to address healthcare disparities has been incorporated. We discussed how expanded roles and reimbursement models could enable pharmacists to deliver advanced preventive and counseling services, thus improving access and outcomes in underserved areas. - Implications for Training and Education
We included a discussion on the importance of sustained and specialized training in tobacco treatment for pharmacists, particularly in rural and underserved areas. We highlighted evidence from educational interventions and emphasized the need for integrating smoking cessation modules into pharmacy curricula and continuing education programs.
We believe these revisions have significantly improved the manuscript by addressing your concerns and providing a more robust analysis. Thank you again for your thoughtful comments, which have been invaluable in refining our work.
Reviewer 3 Report
Comments and Suggestions for Authors
Authors used a descriptive study to “assess patients’ interest in engaging in pharmacist-led cessation programs” (L21). However, this article has not fully answered some of the questions due to insufficient description.
First, authors suggest “Initially, a $5 gift card was provided for survey completion, however to enhance participation, this was later replaced with gift card raffle awards in the amount of $10, $50, and $100.” (P70), but this change of selection of participants may lead to change of characteristics as well as selection bias. Authors should check differences of the characteristics based on the difference of participation, and appropriately add limitations.
Second, authors suggest “did not differ by gender (p=0.79) or age (p=0.51).” (L147), but they do not explain how to calculate “p” as well as the statistical method in method section. It is difficult for readers to understand what authors did in this manuscript. Authors should add explanation regarding statistical analyses in method section.
Minor comments
L: “tobacco/ vaping” may be “tobacco / vaping”.
Author Response
Thank you for your statistical questions, which identify areas to strengthen the Methods and Results sections of our manuscript.
- Potential differences between individuals who received a $5 gift card versus those who were entered into a gift card raffle.
Data were stratified based on the date on which the incentives were changed from a $5 gift card to a gift card raffle. These were approximately evenly divided, with 28 tobacco users receiving a $5 gift card and 29 tobacco users being entered into the raffle. There were no differences between these groups based on gender, age, or number of cigarettes smoked per day. Analyses for race/ethnicity had very small cell sizes, therefore no analysis was conducted. We have chosen not to present these comparisons in the Results section, but we do comment on it in the Discussion, with this language:
To enhance participation, the method of incentives was changed mid-study (49% received a gift card; 51% were entered into a raffle), and this might have introduced bias, although no statistically significant differences were identified for key sociodemographic variables or the number of cigarettes smoked per day (data not shown).
- Statistical Analysis Approach
We have added information to the Methods section to specify the types of comparative tests that we conducted to produce the p-values (Chi-squared tests for categorical variables; t-tests for continuous variables). Thank you for noticing this oversight!
Reviewer 4 Report
Comments and Suggestions for Authors
In the present manuscript, the authors explored smoking behavior and motivations to quit among patients at a Federally Qualified Health Center clinic in rural Wyoming, estimated the prevalence of tobacco-related interventions by clinic staff and characterized patients’ interest in establishing smoking cessation care with a pharmacist. In my opinion, despite the many limitations described by the authors in the article, the work is relatively interesting as it reveals interest among rural Wyoming patients in pharmacist-led smoking cessation services. The results suggest an opportunity for pharmacists to play a key role in supporting cessation efforts.
The introduction provides sufficient background and all references are appropriate and relevant to the research.
The materials and methods are well described; however, perhaps the authors could include the questionnaire used as supplementary material.
The results are clear and well structured. However, the authors should indicate whether there are significant differences between the results obtained and the characteristic studied such as gender, age, race/ethnicity, and frequency of seeing primary care provider.
The conclusions are consistent with the evidence and arguments presented.
Author Response
Thank you for your positive comments regarding the importance of this work in advancing the role of pharmacy in Wyoming (and elsewhere). Two requests were made to improve our manuscript, and our responses are provided below.
- Include the Study Survey
This is now included as a supplemental file.
- Comparative Analyses
The reviewer suggests that additional comparative statistics be presented for differences between the results obtained and characteristics such as gender, age, race/ethnicity, and frequency of seeing a primary care provider. While we do believe that these comparisons would be interesting, we have a small sample size that is not conducive to extensive testing due to small cell sizes and limited degrees of freedom. Additionally, we began this study with no a priori hypotheses. We do, however, believe that these types of analyses are important and should be addressed in a larger study. As such, we have added this in the Discussion section as an area for future research.
Round 2
Reviewer 2 Report
Comments and Suggestions for Authors
Dear Authors, I really appreciated your improvements to the paper.
Author Response
Thank you for your thoughtful feedback and constructive suggestions, which were instrumental in helping us enhance the quality of our paper. We greatly appreciate the time and effort you dedicated to reviewing our work.
Reviewer 3 Report
Comments and Suggestions for Authors
Authors revised the manuscript, but I have some of minor comments.
Minor comments
L73: “models.Toward” may be “models. Toward”.
L293: Authors add the descriptions “”, but they should add the numbers of participants as well as their percentages.
Author Response
Thank you for your careful review and helpful comments. We have corrected the spacing issue on line 73 and added the numbers of participants along with their percentages as suggested on line 293. We appreciate your attention to detail and the opportunity to improve our manuscript.